# Claudins and Gastric Cancer: An Overview

**DOI:** 10.3390/cancers14020290

**Published:** 2022-01-07

**Authors:** Itaru Hashimoto, Takashi Oshima

**Affiliations:** 1Department of Gastrointestinal Surgery, Kanagawa Cancer Center, Yokohama 241-8515, Japan; i-hashimoto@kcch.jp; 2Department of Surgery, Yokohama City University, Yokohama 236-0004, Japan

**Keywords:** gastric cancer, metastasis, claudin, CLDN18

## Abstract

**Simple Summary:**

Gastric cancer (GC) is one of the most common cancers and the third leading cause of cancer deaths worldwide, with a high frequency of recurrence and metastasis, and a poor prognosis. This review presents novel biological and clinical significance of claudin (CLDN) expression in GC, especially CLDN18, and clinical trials centered around CLDN18.2. It also presents new findings for other CLDNs.

**Abstract:**

Despite recent improvements in diagnostic ability and treatment strategies, advanced gastric cancer (GC) has a high frequency of recurrence and metastasis, with poor prognosis. To improve the treatment results of GC, the search for new treatment targets from proteins related to epithelial–mesenchymal transition (EMT) and cell–cell adhesion is currently being conducted. EMT plays an important role in cancer metastasis and is initiated by the loss of cell–cell adhesion, such as tight junctions (TJs), adherens junctions, desmosomes, and gap junctions. Among these, claudins (CLDNs) are highly expressed in some cancers, including GC. Abnormal expression of CLDN1, CLDN2, CLDN3, CLDN4, CLDN6, CLDN7, CLDN10, CLDN11, CLDN14, CLDN17, CLDN18, and CLDN23 have been reported. Among these, CLDN18 is of particular interest. In The Cancer Genome Atlas, GC was classified into four new molecular subtypes, and *CLDN18*–*ARHGAP* fusion was observed in the genomically stable type. An anti-CLDN18.2 antibody drug was recently developed as a therapeutic drug for GC, and the results of clinical trials are highly predictable. Thus, CLDNs are highly expressed in GC as TJs and are expected targets for new antibody drugs. Herein, we review the literature on CLDNs, focusing on CLDN18 in GC.

## 1. Introduction

Gastric cancer (GC) is one of the most common cancers and the third leading cause of cancer-related deaths worldwide, with a high incidence in East Asia [1]. Despite advances in diagnostic equipment and treatment strategies, GC has a poor prognosis, with recurrence and metastasis. Common metastatic sites for GC include the peritoneum, lymph nodes, liver, lungs, and bones [2,3].

The first step in cancer metastasis is local infiltration into the surrounding tumor-related stroma and normal tissue. In this metastatic process, epithelial–mesenchymal transition (EMT) is an important step for cancer cells to acquire a metastatic phenotype in the state of mesenchymal cells [4]. The first stage of EMT is the breakdown of contact between epithelial cells, such as tight junctions (TJs), adherens junctions, desmosomes, and gap junctions, eventually resulting in the loss of cell polarity, cytoskeletal reorganization, and cells. Among these contacts, TJs comprise endogenous membrane proteins, such as claudin (CLDN) and occludin, and cytoplasmic proteins zonula occludens 1 (ZO1), ZO2, and ZO3, which are transmembrane proteins. They link the actin cytoskeleton and signaling proteins [4,5,6]. In general, CLDNs play an essential role in scaffolding for cell–cell adhesion and migration. The expression of CLDN has been reported in several types of cancer, including GC, and has recently attracted attention as a new therapeutic target [7]. In 2014, The Cancer Genome Atlas (TCGA) classified GC into four subtypes: Epstein–Barr virus (EBV)-positive, microsatellite instability, genomic stability (GS), and chromosomal instability. Among these, it was reported that the characteristics of GS are rich in mutations in the fusion of *CLDN18*–*ARHGAP*, in addition to the diffuse histological type and *RHOA* [8]. Furthermore, the efficacy of anti-CLDN18 antibody drugs in patients with GC has been reported, and clinical trials are currently underway [9,10].

In this review, we add new findings to literature on CLDNs, with a focus on CLDN18 in GC, and highlight its potential for research as well as the clinical application of CLDNs.

## 2. General Information of CLDNs

CLDNs are a family of at least 27 transmembrane proteins [11,12] and were first reported by Tsukita et al. [13]. CLDNs are classified into classic and non-classic types according to their difference in sequence [11,14]. Classic types include CLDNs1–10, CLDN14, CLDN15, CLDN17, CLDN19, and non-classic types include CLDNs11–13, CLDN16, CLDN18, and CLDNs20–24 [14]. CLDNs are 20–34 kDa in size [14] and are composed of the N-terminal region of the cytoplasm, two extracellular loops, four transmembrane domains, and the C-terminal tail of the cytoplasm [7,14] (Figure 1). The large extracellular loops of CLDNs have charged amino acids and form charge-selective channels that regulate the ion selectivity of neighboring cells [15]. The two cysteines of CLDNs may form internal disulfide bonds to stabilize protein conformation [16], and the shorter second extracellular loop is folded with a helix-turn-helix motif. It may be involved in dimer formation between CLDNs on opposite cell membranes through hydrophobic interactions between conserved aromatic residues [17]. The C-terminus of the CLDN protein exhibits diversity in sequence and length. Most CLDN family members provide scaffolding for migration, cell adhesion, matrix remodeling, and proliferation. It forms a C-terminal PDZ-binding motif that interacts with ZO1, ZO2, ZO3, and multi-PDZ domain protein 1 (MUPP1) [18,19,20]. This region also contains amino acid residues associated with post-translational modifications such as serine–threonine phosphorylation, tyrosine phosphorylation, SUMOylation, and palmitoylation, all of which affect the localization and function of CLDNs [21,22,23,24,25].

Furthermore, the phosphorylation of the C-terminal tail of CLDNs interacts with several major kinases. CLDN3 is phosphorylated by Protein Kinase A, which induces disruption of the TJs [26]. CLDN5 has been reported to increase the permeability of Palauan endothelium through phosphorylation via protein kinases Cα and ζ [27]. CLDNs1–4 are phosphorylated by the threonine/serine kinase WNK4 [28], and CLDN4 is phosphorylated at the PDZ-binding C-terminus by the receptor tyrosine kinase EphA2, which reduces the binding to ZO1 and is intercellular and promotes contact withdrawal [29]. The phosphorylation of CLDNs as described above may downregulate TJ intensity [26,30].

From a structural and functional view, CLDNs generally localize to the apical region of the cell membrane and form a TJ complex for cell–cell adhesion, maintenance of cell polarity, and selective paracellular permeability, and play an important role in barrier function [6,31,32,33,34,35]. As loss of the cell–cell adhesion complex is associated with increased EMT in cancer, phosphorylation and reduction of CLDNs may promote metastasis and infiltration. Alternatively, overexpression of CLDNs has also been reported to increase aberrant localization and function in gastric, lung, prostate, ovarian, colorectal and breast cancers, promoting metastasis and progression [7].

Thus, CLDNs are generally highly expressed in cancer tissues depending on the type of cancer and the type of CLDN, although their reduction or loss of function due to phosphorylation promotes EMT and cancer metastasis and infiltration.

## 3. CLDN Expression in GC

The expression of several CLDNs has been described in GC, including their functions, molecules involved, and clinical significances (Table 1). The following section summarizes new insights and deeper considerations for each CLDN’s expression in GC.

### 3.1. CLDN18

The CLDN18 protein is the most studied among the CLDNs, since it is specifically expressed in stomach and GC tissue, compared to other tissues or cancers, which makes it a potential therapeutic target. CLDN18 was first identified in 2001 as a downstream target gene of the T/EBP/NKX2.1 homeodomain transcription factor [36]. The human *CLDN18* gene has two splice variants, which encode two protein isoforms, CLDN18 splice variant 1 (CLDN18.1) and CLDN18 splice variant 2 (CLDN18.2) [37]. CLDN18.1 is specifically expressed in normal and cancerous lung tissues, and CLDN18.2 is expressed in normal gastric tissues and in tissues of gastric, pancreatic, esophageal, and lung cancers [37]. In the normal gastric mucosa, CLDN18 is found on the surface of the lobular epithelium, immature cells, and glandular epithelium. In intestinal metaplasia, CLDN18 is absent in metaplastic epithelium [38].

**Table 1 cancers-14-00290-t001:** Claudin’s functions, signaling molecules involved, and clinical significances in GC.

Type of CLDN	Main Functions, Signaling Molecules Involved, and Clinical Significances in GC	References
CLDN1	- correlation with tumor infiltration and metastasis - Akt, Src, and NF-κB signaling pathway - poor prognostic factor	[39,40] [41,42] [40,43]
CLDN2	- CDX2-dependent targeting relationship with CagA produced from *H. pylori*	[44]
CLDN3	- correlation with lymph node metastasis - important immunosuppressive regulator	[45] [46]
CLDN4	- correlation with lymph node metastasis - promoting EMT and infiltration of MMP-2 and MMP-9	[47] [48,49,50]
CLDN6	- cell proliferation and migration/infiltration with YAP1 - high expression was positively correlated with decreased OS	[51] [51,52]
CLDN7	- proliferation in a CagA-and β-catenin-dependent manner - poor prognostic factor	[53] [54]
CLDN10	- association with metastasis and proliferation	[55,56]
CLDN11	- correlation with H. pylori infection and Borrmann classification, not with lymph node metastasis and TNM stage	[57]
CLDN14	- correlation with lymph node metastasis	[58]
CLDN17	- correlation with lymph node metastasis	[58]
CLDN18	- correlation with metastasis (lymph node, peritoneal, bone, and liver metastasis) - poor prognostic factor - Wnt, β-catenin, CD44, EFNB/ EPHB receptor signals, and HIPPO signals - *CLDN18*-*ARHGAP* fusion in genomically stable type - therapeutic target of Claudiximab (IMAB362, Zolbetuximab)	[59,60] [61,62,63,64,65,66,67,68] [54,69,70] [8] [10,71,72,73]
CLDN23	- poor prognostic factor	[57]

Downregulation of CLDN18 has been reported to occur during intestinal metaplasia [74,75] in the Correa cascade, a multistep and multifactorial process of gastric carcinogenesis [69,70]. Notably, the expression of CLDN18 in GCs has different biological functions depending on whether it is up- or downregulated; CLDN18 is also highly expressed in normal gastric tissues, and downregulation of this expression was detected in 57.5% of GCs, especially in 73.7% of intestinal phenotypes [69]. Similarly, CLDN18 was downregulated in GC compared to the normal mucosa of the surrounding stomach and intestinal metaplasia [54,76].

In the relationship between CLDN18 expression and clinicopathological factors in GC, CLDN18 expression was significantly lower in patients with peritoneal metastasis (PM) than those without PM (*p* = 0.01). Meanwhile, CLDN18 expression was significantly higher in patients with bone metastasis than in those without bone metastasis (*p* = 0.01) [59]. Another IHC study showed that CLDN18 expression correlated with lymph node metastasis (*p* = 0.04), high stage disease (III, IV) (*p* = 0.019), and lower incidence of liver metastases (*p* = 0.009) [60]. In addition, a vital relationship existed between the decreased expression of CLDN18 and perineural invasion [54].

In tissues of early-stage GC removed via endoscopic mucosal resection or endoscopic submucosal resection, the Ki-67 labeling index at the invasive front inversely correlated with the CLDN18 expression level, suggesting that a decrease in CLDN18 expression promotes cancer invasion in early-stage GC [76]. Furthermore, concerning the relationship between CLDN18 expression and survival, patients without CLDN18 expression reported to have shorter overall survival (OS) than those with CLDN18 expression [54,69,70].

To elucidate the biological significance and carcinogenic mechanism of reduced CLDN18 expression in GC, studies were conducted on GC cell lines in which CLDN18 was knocked down by siRNA and knocked out in mice [76,77]. GC cells knocked down by CLDN18 promoted gastric cancer cell proliferation and infiltration compared to the controls [76]. In CLDN18 knockout mice, CLDN18 was reported to be localized to TJs at the base rather than at the apex of gastric epithelial cells [77]. Furthermore, in CLDN18 knockout mice, gastric mucosal atrophy and convulsive polypeptide expression alteration (SPEM) occur after paracellular H^+^ ion leakage and parietal cell death, and SPEM was suggested as the origin of cancer stem cells [78].

However, it has also been reported that CLDN18 knockout mice do not progress to cancer while maintaining atrophic gastritis and SPEM status [79]. In other mouse models, CLDN18 deficiency occurred in the gastric mucosa of *Helicobacter pylori*-infected mice [77]. In contrast, the inactivation of CLDN18 in mice not infected by *H*. *pylori* promotes the growth of intraepithelial neoplasms that develop into polypoid tumors, β-catenin [61,62], CD44 [63], and EFNB/ EPHB receptor signals [64,65]. HIPPO signals [66,67] and other vital signaling pathways promote cell proliferation, cancer stem cell development, and tumorigenesis. In GC CLDN18 knockout mice, chronic active gastritis developed at middle age (>40 weeks), and the expression of CCL28, a chemokine with lymphocyte chemo-ventilation activity, was observed. At old age (60 weeks or older), 20–30% of these mice develop gastric tumors, and CXCL5 is expressed. These are multifunctional cytokines with neutrophil-attracting, angiogenic, and EMT-inducing effects. During this process, SPEM cells developed, and the expression of CD44-variants, TLR2, and CXCL5 increased. Some features of gastric tumorigenesis in CLDN18 knockdown mice resemble human carcinogenesis associated with *H*. *pylori* infection. Wnt1 overexpression in transgenic CLDN18 knockout mice promotes gastric tumorigenesis. This indicates that when gastritis is induced by CLDN18 deficiency, Wnt-dependent gastric tumorigenesis may be triggered [68].

While CLDN18 downregulation is crucial for GC development, proliferation, infiltration, and EMT, positive expression or upregulation of CLDN18 also plays a biologically important role in GC. In immunohistochemistry (IHC) studies, CLDN18 expression was found in 42.2% of GCs and correlated with the mucin phenotype, EBV status, integrin αvβ5, EpCAM extracellular domain EpEX, and lysozymes [80]. Bioinformatics analysis of *CLDN18* expression in GC patients using multiple public databases revealed that *CLDN18* expression was higher in the EBV-positive group than in other groups [81]. Tissue microarray analysis showed high membranous CLDN18 expression in 29.4% of primary cases and 34.1% of metastatic cases. Positive expression of membrane-type CLDN18 was significantly associated with the Lauren diffuse type (*p* = 0.009) and EBV-related cancers (*p* < 0.001) [82]. In other IHC studies, CLDN18 was frequently expressed at the primary site, GC. In the metastatic cohort, 88% of GCs were positive for CLDN18 [83].

The TCGA Research Network [8] revealed that the *CLDN18*–*ARHGAP26/6* translocation was enriched in genomically stable tumors, which was also confirmed in diffuse GC [84]. In the East Asian GC population, *CLDN18*–*ARHGAP26/6* was detected in 3% of all cases [85]. In other aspects, this unique fusion gene is associated with younger patients and invasive diseases, including lymph node and distant metastases [86,87]. Histologically, signet ring cell carcinoma (SRCC), which is based on microscopic characteristics according to the classification by the World Health Organization, belongs to the diffuse type [88]. The whole-genome sequence of 32 SRCC samples from the eastern Chinese population confirmed that the frequency of *CLDN18*–*ARHGAP26/6* fusion was 25%. In the validation cohort, patients with *CLDN18*–*ARHGAP26/6* fusion had poorer survival than those without fusion. Moreover, they did not benefit from oxaliplatin/fluoropyrimidine-based chemotherapy [89]. This is consistent with the chemical drug resistance acquired in GC cells overexpressing *CLDN18*–*ARHGAP26*.

The subtype of gastric intestinal adenocarcinoma with anastomotic glands has been reported to be frequently associated with poorly differentiated adenocarcinoma components [90,91]. In recent years, *RHOA* mutations and *CLDN18*–*ARHGAP* fusions are typically present in adenocarcinomas with anastomotic glands via next-generation sequencer and reverse transcription-PCR [92]. This characteristic chimeric protein CLDN18–ARHGAP in GC may also be associated with carcinogenesis through functional changes, such as the disappearance of CLDN18 and the acquisition of ARHGAP function (Figure 2). Furthermore, the carboxyl-terminal PDZ-binding motif of CLDN18 interacts with actin-regulating proteins such as RhoA, cadherin, and integrins [93]. Therefore, loss of the carboxyl-terminal domain of CLDN upon fusion with ARHGAP may prevent actin-regulated proteins from binding to the junctional complex and loosen cell–cell and/or cell–matrix binding. This aberrant ectopic activity of ARHGAP may be involved in carcinogenesis, just as the constitutive inactivation of RhoA leads to the acquisition of a carcinogenic phenotype [94,95]. Alternatively, there are reports that mutations in *RHOA* are gain-of-function [84,96], and further studies are needed concerning the mechanism and effects of this unique fusion of *CLDN18*–*ARHGAP* in GC. Several cancer cell lines stably expressing *CLDN18*–*ARHGAP26* showed a dramatic loss of the epithelial phenotype and long protrusions showing EMT. Fusion-positive cell lines showed loss of barrier properties, reduced cell–cell and cell–extracellular matrix adhesion, delayed wound healing, RhoA inhibition, and acquisition of invasiveness. Thus, *CLDN18*–*ARHGAP26* contributes to epithelial degradation, possibly causing H^+^ leakage in the stomach, inducing inflammation, and promoting infiltration [85]. In addition to the *CLDN18* (exon5)-*ARHGAP26* (exon12 or exon10) or *ARHGAP6* (exon2), fusion patterns reported in TCGA cohort [8] and other reports [87,92] have shown *CLDN18* (exon5)–*ARHGAP26* (exon12 or exon10) or *ARHGAP6* (exon2). There are rare cases of *CLDN18* (exon4)–*ARHGAP26* (exon11) [89], *CLDN18* (exon5)–*ARHGAP10* (exon8), and *CLDN18* (exon5)–*ARHGAP42* (exon7) [87]. Notably, all these *CLDN18*–*ARHGAP* fusions share a common RhoGAP domain after gene translocation and are thought to promote carcinogenesis and cancer progression by inactivating Rho.

CLDN 18.2 has been reported as a target for therapeutic antibodies [36,37,97,98,99]. In normal gastric tissue, CLDN18.2 is contained in TJ supramolecular complexes of gastric mucosal cells, and the epitope of CLDN18.2 has little access to intravenous antibodies [36,37,100]. However, the loss of cell polarity associated with malignancy exposes the epitope of CLDN18.2, making it accessible to bound antibodies, and CLDN18.2, which is maintained in GC and gastric metastases [37,54,101]. Claudiximab (IMAB362, Zolbetuximab) is a novel chimeric IgG1 antibody that is highly specific for CLDN18.2. Claudiximab is derived from a mouse monoclonal antibody and is chimeric with the human IgG1 constant region for clinical application (Table 2). This novel antibody binds to CLDN18.2 on the surface of cancer cells and stimulates cells and immune effectors that promote antibody-dependent cellular cytotoxicity and complement-dependent cellular cytotoxicity [71]. Moreover, it can induce apoptosis and suppress cell proliferation. Combination therapy with Claudiximab and chemotherapy may promote T-cell infiltration and induce inflammatory cytokines [32].

CLDN18.2-positive metastatic, refractory patients were enrolled in a phase I single-dose escalation study of IMAB362 with gastric or esophagogastric junction cancer. Three patients were enrolled in each of the five dose cohorts. In this study, IMAB362 was generally well tolerated at all dose levels; however, the most common adverse event was nausea and vomiting. No dose-limiting toxicity was observed within four weeks. Most patients progressed 4–5 weeks after a single intravenous dose of IMAB362; however, one patient in the 600 mg/m^2^ group was stable for approximately 2 months after administration. Based on the pharmacokinetic profile and preclinical dose–response data obtained in this study, a dose of 300–600 mg/m^2^ was recommended in a phase II multidrug study [71].

Another phase I study, PILOT, included IMAB362 plus zoledronic acid (ZA) and a low-to-medium dose interleukin 2 (IL-2) in patients with advanced GC and esophagogastric junction adenocarcinoma. The safety and efficacy of this combination was evaluated (NCT01671774) [72]. Twenty-eight patients were enrolled, with IMAB362 and ZA in Group 1; IMAB362, ZA, and IL-2 (1 × 10^6^ IU, sc) in Group 2; IMAB362, ZA, and IL-2 in Group 3 (3 × 10^6^ IU, sc); and IMAB362 alone in Group 4. Among the 20 patients, 11 gained disease control (one with an unidentified partial response and 10 with disease stability). The median progression-free survival was 12.7 weeks, and the median OS was 40 weeks. Grade 1–3 nausea and vomiting were predominant adverse events associated with IMAB362. The PILOT trial revealed that the combination therapy of IMAB and ZA/IL-2 had antitumor effects and was well tolerated.

The multicenter, phase IIa “MONO” study [9], which follows the phase 1 study “PILOT”, was effective in administering IMAB362 multiple times as monotherapy in patients with GC or lower esophageal cancer [72]. This was done with the aim of establishing effectiveness and safety. Forty-four patients participated in this study, and data on the antitumor effect of 43 patients were obtained, with an objective response rate (ORR) of 4 (9%) and a clinical efficacy rate (ORR + SD) of 23%. Adverse events occurred in 81.5% (44/54) of patients, with nausea (63%), vomiting (57%), and fatigue (43%). Notably, patients who underwent total gastrectomy had a lower frequency of severe gastrointestinal adverse events (AEs) than those who did not.

The randomized phase IIb study of IMAB362 plus anticancer drug, FAST study, were first-line treatment for patients with advanced gastric–esophageal junction or esophageal adenocarcinoma expressing CLDN 18.2, compared to epirubicin + oxaliplatin + capecitabine (EOX). This study evaluated the efficacy, safety, and tolerability of IMAB362 in combination with EOX [10]. Patients received EOX (*n* = 84) every 3 weeks or received IMB362 + EOX (*n* = 77) as an initial treatment. After enrollment, IMAB362 + EOX increased to 1000 mg/m^2^ (*n* = 85) as an exploratory case group. Progression-free survival (hazard ratio (HR) = 0.44, 95% confidence interval (CI) = 0.29–0.67, *p* < 0.0005) and OS (HR = 0.55, 95% CI = 0.39–0.77, *p* < 0.0005) in the entire population were extremely good with IMAB362 + EOX (arm2) compared to EOX alone (arm1). Most AEs associated with IMAB362 + EOX were grades 1–2. AEs with grade ≥3 did not show an overall increase compared to EOX alone. These results showed that treatment with IMAB362 targeting CLDN18.2 for patients with GC is an effective and safe treatment. Alternatively, a phase 3 trial, GLOW (NCT03653507), comparing IMAB362 + CAPOX with placebo + CAPOX is underway as a first-line treatment for patients with gastroesophageal junction adenocarcinoma [73].

Moreover, recently, new antibody drugs that can be expected to have additional therapeutic effects have been developed by modifying existing antibody drugs. This study succeeded in developing a humanized CLDN18.2 specific single-chain fragment variable (scFv) [102]. Subsequently, CLDN18.2 specific CAR T cells were developed using scFv as a targeting component. CLDN 18.2-specific CAR T cells containing the CD28 costimulatory domain suppressed tumor growth in xenograft mouse models of cancer cell lines. When CAR T cells were administered to a xenograft (PDX) model from a CLDN18.2-positive patient with GC, a partial or complete tumor response was observed. CAR T cells bind well in vivo and efficiently infiltrate tumor tissues. CLDN18.2 CAR T cells could lyse target cancer cells expressing mouse CLDN18.2; however, it had no apparent adverse effects on normal organs, including gastric tissue. In addition, antibody drug conjugates (ADCs) and CD3 bispecific antibodies are being developed as new targeted antibody therapies. Zhu et al. conducted efficacy and preliminary toxicity studies of CLDN18.2 target antibodies via ADC and CD3-bispecific antibodies and their potential therapeutic molecules, with anti-hCLDN18.2 ADC, CD3-bispecific, and diabody. KATO-III/hCLDN18.2 showed in vitro cytotoxicity and suppressed tumor growth in xenograft tumors derived from gastric patients. In a preliminary assessment of tolerability, anti-CLDN18.2 diabodies showed no toxicity in the stomach of NSG mice 4 weeks after dosing [103]. Given these findings, targeting CLDN18.2 with ADC or a bispecific modality may be a new therapeutic approach for the treatment of GC. To date, most IMAB362 studies employed IHC (CLAUDETECT 18.2VR kit) to assess CLDN18.2 expression in patients with GC. However, given that 92% of the protein sequence of CLDN18.1 is highly consistent with CLDN18.2, the search for more specific CLDN18.2 antibodies remains a major challenge. As a new approach, the CLDN18.2 molecular beacon (MB) with a stem-loop hairpin structure was reported for the detection of CLDN18.2 in blood samples. This MB rapidly recognized the RNA of CLDN18.2 [104] and was successfully applied to the circulating tumor cell (CTC) assay. This new method of detecting CLDN18.2 RNA in CTCs may be a new approach for identifying potential patients with CLDN 18.2 target drugs.

### 3.2. CLDN1

CLDN1 is most abundant in squamous epithelium [105]; however, it has also been reported to be overexpressed in GC tissues [39,40]. Overexpression in GC has also been reported to correlate with tumor infiltration and metastasis and as a poor survival factor [40,43]. In addition, overexpression of CLDN1 promoted cell aggregation and enhanced anoikis resistance. In contrast, low expression of CLDN1 has been reported to be associated with diffuse-type GC [106,107]. Huang et al. reported that CLDN1 deficiency inhibits cell migration, infiltration, and colonization in vitro and inhibits tumorigenesis and metastasis in vivo. Elevated β-catenin in CLDN1 knockdown cells restored cell aggregation and anoikis resistance and reactivated the Akt and Src signaling pathways [41]. CLDN1 was knocked down in GC cells, and microarray analysis was used to identify 245 genes with altered expression levels. According to the pathway analysis of these genes, the top-ranked molecular and cellular functions were MMP7, TNF-SF10, TGFBR1, and CCL2. TNF-α and NF-κB were ranked as upstream regulators associated with CLDN1. Knockdown of *CLDN1* in response to this result suppressed the expression of TNF-α mRNA [42].

### 3.3. CLDN2

CLDN2 is expressed in the gallbladder, kidney, and gastrointestinal tract [105]. In the large intestine, it has been reported to be selectively increased in colorectal cancer and considered useful as a tumor marker and a target for the treatment of colorectal cancer [108]. CLDN2 expression in breast cancer tissue was reduced compared to that in normal breast tissue, and low CLDN2 expression was associated with lymph node metastasis and worsening clinical stage of breast cancer [109]. Similarly, the expression level of CLDN2 in the stomach was reportedly higher in GC tissues than in normal tissues [110,111] and gradually increased during the multistep expression process in GC [110]. In contrast, CLDN2 is reportedly less expressed in GC than in the adjacent normal mucosa [112,113]. Regarding the relationship between CLDN2 expression and clinical pathological factors, no significant association between the two was observed [113]. Song et al. revealed that CagA produced from *H*. *pylori* promotes the infiltration of GC cell lines through a CDX2-dependent targeting relationship with CLDN2 [44]. CDX2 increases CLDN2 expression at the transcriptional and translational levels, and TJs in GC cells infected with CagA-positive wild-type *H*. *pylori* are more severely disrupted than CagA-negative allogeneic variants. It has been reported that gaps between cells widen, contact is impeded and scattered, and cell migration is greatly enhanced [44].

### 3.4. CLDN3

CLDN3 is expressed in the gastrointestinal tract, including the stomach and in several tissues, including glandular tissue [105]. The expression of CLDN3 in GC tissue obtained from samples of endoscopic gastric mucosal resection and endoscopic submucosal resection was higher than that in the adjacent normal mucosa; however, it was lower in the anterior surface of GC submucosal invasion [114]. CLDN3 was lower in patients with positive lymphatic invasion, advanced tumor depth [115], and lower TNM stage [116]. Conversely, high CLDN3 expression was related to lymph node metastasis [45]. CLDN3 was strongly expressed in most gastrointestinal adenocarcinomas; however, less frequently in diffuse GC [106], and high expression of CLDN3 in the intestinal form was associated with good patient survival [107]. Currently, CLDN3 has been found highly expressed in immunologically responsive tumors and negatively correlated with GC CD8^+^ T cells. Overexpression of CLDN3 in GC cells suppressed the expression of MHC-I and CXCL9, chemokines essential for the recruitment of CD8^+^ T lymphocytes [46]. In future, it is expected to be a biomarker candidate for GC, including the prediction of the effect of immune checkpoint inhibitors.

### 3.5. CLDN4

CLDN4 is expressed on the cell membranes of various tissues and is most abundantly expressed in the gastrointestinal tract [105]. Abnormal expression of CLDN4 was detected in GC and its precursor lesions [117,118]. In a meta-analysis, CLDN4 expression was associated with increased pT classification, tumor size, and lymph node metastasis in patients with GC [47]. There are some contradictory reports that CLDN4 upregulation [48] or downregulation [49] may induce metastasis through the promotion of EMT. Overexpression of CLDN4 degrades extracellular matrix components and ultimately promotes the infiltration and motility of cancer cell metalloprotease-2 (MMP-2) and metalloprotease-9 (MMP-9) [50]. Conversely, given that CLDN4 plays an important role in the formation of TJs, weakening of TJs due to abnormal expression of CLDN4 reduces the stability of cell–cell adhesion and thus promotes detachment. Kwon et al. reported that overexpression of CLDN4 was associated with clinicopathological factors and inhibited GC cell migration and infiltration. High expression of CLDN4 was shown to enhance the barrier function of TJs, and strongly correlates with DNA hypomethylation in GC. They reported that the loss of inhibitory histone methylation and acquisition of active histone modifications were associated with the overexpression of CLDN4 in GC cells [119]. Recently, studies have begun to characterize the potential regulatory effects of non-coding RNAs such as microRNAs (miRNAs) and long non-coding RNAs (lncRNAs) on GC [120,121]. lncRNA-KRTAP5-AS1 and lncRNA-TUBB2A may act as competing endogenous RNAs that affect CLDN4 function [122]. Ma et al. showed that EZH2/GJA1 and miR-1207-5p/CLDN4 mediated by BTEB2-activated-TSPEAR-AS2 play important roles in the progression of GC [123]. Collectively, the expression of CLDN4 by lncRNA may be a potential biomarker or therapeutic target for GC. Anti-human claudin-4 antibody (4D3) has been developed as a targeted therapy for CLDN4 [124]. Nishiguchi et al. reported that 4D3 may increase sensitivity to chemotherapy by inducing structural disruption of TJs of CLDN4 expressed in GC [125].

### 3.6. CLDN6

CLDN6 is highly expressed in the placenta and testes [105]. Several reports have focused on the relationship between CLDN6 expression and survival in GC tissues. Expression of CLDN6 mRNA and protein in GC tissue has been reported to be lower than that in the nearby normal mucosa [113,126]. In contrast, CLDN6 was identified as an overexpressed gene in GC tumors compared to adjacent normal tissues, and its high expression was positively correlated with decreased OS in patients with GC [51,52]. In addition, there was no significant relationship between *CLDN6* expression and metastatic state (lymph node and organ metastasis) using the TCGA dataset [51]. Histologically, this expression is associated with intestinal GC [51] and AFP-producing gastric adenocarcinoma [127]. Knockdown of CLDN6 in GC cells has been suggested to suppress cell proliferation and migration/infiltration by partially suppressing the transcription of YAP1 and its downstream transcription targets [51]. Similarly, Yu et al. reported that CLDN6 reduced the phosphorylation of YAP1 by acting on LATS1/2, and increased the nuclear translocation of YAP1 to activate downstream target genes and EMT [52].

### 3.7. CLDN7

CLDN7 mRNA is highly expressed in the stomach, and its protein is expressed in most acinar cells [105]. The relationship between CLDN7 expression in GC and clinicopathological factors negatively correlated in patients with diffuse type and lymphatic invasion [54]. There was no significant relationship between CLDN7 expression and metastatic state (lymph node and organ metastasis) [54]. Survival analysis showed that patients with CLDN7 had shorter OS than those without CLDN7 [54]. It has been reported that *H*. *pylori* infection in GC alters the localization of CLDN7 and increases proliferation in a CagA-and β-catenin-dependent manner [53]. In GC cells, *H*. *pylori* suppresses CLDN 7 expression via β-catenin and Snail, whereas in *H*. *pylori*-infected individuals, Snail expression is increased and CLDN 7 levels are decreased [53]. It has been suggested that *H*. *pylori* infection during carcinogenesis is associated with the CLDN7 protein.

### 3.8. CLDN10

CLDN10 is highly expressed in the cell membranes of several tissues, especially in the renal tubules, exocrine pancreas, gallbladder, and salivary glands [105]. Abnormal expression of CLDN10 in GC may be associated with cancer progression. In some cancer types, the relationship between CLDN10 overexpression and clinicopathological factors was associated with metastasis [55], cell proliferation, and infiltration capacity [56]. In contrast, the expression of CLDN10 in laryngeal cancer tissue and normal tissue was not different in any pathological factor [128]. To date, very few studies have been conducted on the expression of CLDN10 in GC. Gao et al. reported that the positive expression rate of CLDN10 in GC tissue and adjacent non-tumor tissue was 24% and 72%, respectively, and that CLDN10 and E-cadherin were simultaneously expressed in GC [58]. In this regard, the disappearance of CLDN10 and E-cadherin may contribute to the loosening of cell adhesion in GC.

### 3.9. CLDN11

CLDN11 is abundantly expressed in the membranes of Sertoli cells in the testes and oligodendrocytes of the CNS [105]. In cancer, CLDN11 interacts with alpha1 integrin to regulate oligodendrocyte proliferation and migration [129]. In myeloma and bladder cancer, the loss of CLDN11 may be associated with proliferation, recurrence, and invasiveness [130,131]. CLDN11 was found to be highly expressed in GC tissues compared to the adjacent normal mucosa [113]. In contrast, CLDN11 expression in GC tissues was lower than that in superficial gastritis [57]. The relationship between CLDN11 expression in GC and clinicopathological factors correlated with gender, smoking, alcohol consumption, *H*. *pylori* infection, and Borrmann classification, not with lymph node metastasis and TNM stage.

### 3.10. CLDN14

CLDN14 mRNA is highly expressed, especially in the liver and kidneys [105]. However, low expression of CLDN14 was found in 129/212 (60.8%) patients with primary hepatocellular carcinoma (HCC). Furthermore, it has been reported that low expression of CLDN14 positively correlated with tumor size and clinical stage of HCC and is an independent prognostic factor for poor survival of HCC patients [132]. Furthermore, decreased expression of CLDN14 in HCC cells was associated with increased Wnt/β-catenin signaling. To date, there are limited studies on the expression of CLDN14 in GC; however, evaluation of CLDN14 expression by IHC reported that GC tissues had higher expression compared to proximal normal mucosa and that CLDN14 expression correlated with lymph node metastasis and E-cadherin expression [58].

### 3.11. CLDN17

The RNA of CLDN17 is expressed in the esophagus, salivary glands, tongue, and vagina. Its protein expression is also expressed in the cytoplasm of the oral mucosa and esophagus [105]. The positive expression rate of CLDN17 in GC tissues was lower than that in normal tissues. Low expression of CLDN17 was positively associated with lymph node metastasis and negatively associated with E-cadherin expression [58].

### 3.12. CLDN23

CLDN23 is detected in several tissues and is a prognostic marker for renal and colorectal cancers [105]. Intestinal GC reduced the expression of CLDN23 RNA [133]. In addition, CLDN23 was reduced in cancerous tissues compared to that in the nearby normal mucosa. Clinicopathologically, CLDN23 was not associated with lymph node metastasis and TNM stage. Cox multivariate survival analysis showed that OS was longer when CLDN23 expression was negative in patients with GC [57].

## 4. Conclusions

CLDNs, as TJs, play a central role in the intercellular barrier in normal tissues and are essential for maintaining homeostasis in living organisms. Abnormal expression of CLDNs has been observed in cancers, suggesting that these may induce EMT-mediated carcinogenesis and metastasis. In GC, it has been reported that the expression of various CLDNs is increased or decreased depending on the histological type (intestinal cancer, diffuse cancer) and the stage of cancer (carcinogenesis, metastasis). Among them, CLDN18, a characteristic fusion (*CLDN18*–*ARHGAP*), has been discovered in the development of the anti-CLDN18.2 antibody drug and in GS, which is one of the genomic classifications of GC by TCGA. It is also expected to be a new therapeutic target.

## Figures and Tables

**Figure 1 cancers-14-00290-f001:**
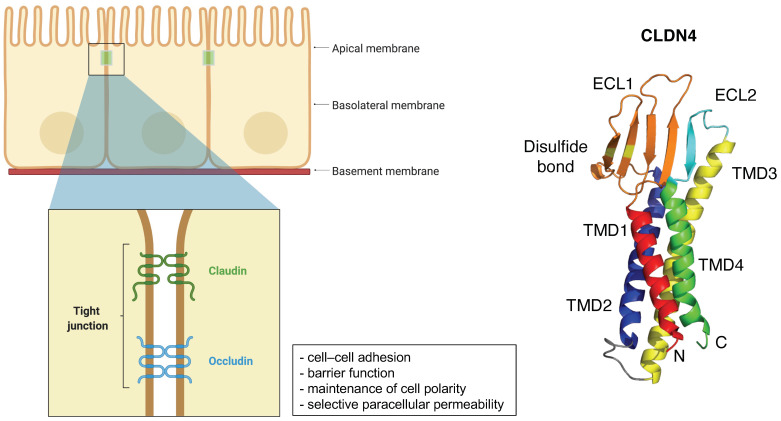
Schematic of the claudin protein located in tight junction and the 3D structures of CLDN4 (PDB: 7KP4). Adapted from “Cell Junction Types”, by BioRender.com (2021). Retrieved from https://app.biorender.com/biorender-templates (accessed on 30 September 2021).

**Figure 2 cancers-14-00290-f002:**
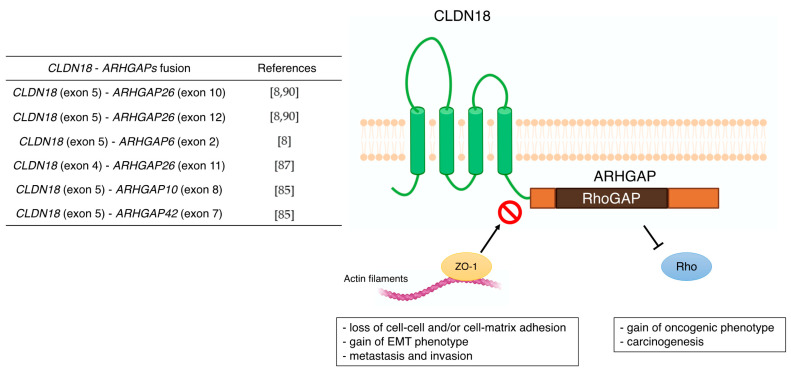
Previous reports of *CLDN18*-*ARHGAPs* and schematic of *CLDN18*-*ARHGAP* protein. Created with BioRender.com.

**Table 2 cancers-14-00290-t002:** Clinical trials associated with IMAB362.

Study Name	NCT Number	Phase	Number of Participants	Design	Response Rate	OS	PFS	Adverse Effects
-	NCT00909025	I	15	Single-dose escalation study evaluating safety and tolerability	-	-	-	Vomiting
PILOT	NCT01671774	I	32	Multiple dose study of IMAB362 with immunomodulation (Zoledronic acid and IL-2)	11 patients had disease control	40 weeks	12.7 weeks	Nausea and vomiting
MONO	NCT01197885	IIa	54	Multiple dose study of IMAB362 as monotherapy	Clinical benefit rate: 23%	-	-	Nausea, vomiting, and fatigue
FAST	NCT01630083	IIb	246	Randomized EOX vs. IMAB362 + EOX, extended with high-dose IMAB362 + EOX	Objective response rate: 25 vs. 39%	8.3 vs. 13.0 months	5.3 vs. 7.5 months	Neutropenia, anemia, weight loss, and vomiting
GLOW	NCT03653507	III	500 (estimated)	Double-blinded, Randomized, IMAB362 plus CAPOX compared with placebo plus CAPOX as first-line treatment of subjects with CLDN 18.2-positive, HER2-negative locally advanced unresectable or metastatic gastric or gastroesophageal junction adenocarcinoma	-	-	-	-

OS, overall survival; PFS, progression-free survival.

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
