# Peer review of "Claudins and Gastric Cancer: An Overview"

_cancers, 2022, doi:10.3390/cancers14020290_

Round 1
Reviewer 1 Report
This is a nicely written review. A few suggestions for improvement:
- Please add to the introduction information about Claudins in general and the general function. Currently, only CLDN18.2 is mentioned in the introduction, but an overview of these molecules would help. Some general information is given in the second paragraph, but a sentence or two would help in the introduction to define these molecules. Something saying they family of proteins generally plays a role in scaffolding for adhesion and migration.
- The sentence in line 77 is not clear to me, could you please reword? Is CLDN3 phosphorylated by protein kinase A? Then what happens? What activity does it go on to do?
- In section 3, perhaps an introductory sentence on why gastric cancer is being focused on would help. Does CLDN18 more important in gastric cancer than other cancers?
- In section 3.1, please provide an overview of CLDN18 in metastasis since the special issue is on metastasis. Only lymph node metastasis where it did not correlate is mentioned so any literature in metastasis in general would help.
- Some of the CLDNs sections do not have a mention of metastasis (CLDN 3, 6, 7, 11, 14, and 23). Please check the literature and TCGA for this or mention it is not known in those sections
Author Response
Response to Reviewer 1 Comments
Point 1: Please add to the introduction information about Claudins in general and the general function. Currently, only CLDN18.2 is mentioned in the introduction, but an overview of these molecules would help. Some general information is given in the second paragraph, but a sentence or two would help in the introduction to define these molecules. Something saying they family of proteins generally plays a role in scaffolding for adhesion and migration.
Response 1: Thank you very much for your constructive suggestion. We agreed with this comment and added a key sentence.
Changes: The second paragraph in introduction section of the manuscript now contains the following sentence: ‘In general, CLDN proteins play an essential role in scaffolding for cell-cell adhesion and migration.’ (line 44-45)
Point 2: The sentence in line 77 is not clear to me, could you please reword? Is CLDN3 phosphorylated by protein kinase A? Then what happens? What activity does it go on to do?
Response 2: Thank you very much for your point out. We revised the sentence and added new information of phosphorelated CLDN3 activity.
Changes: The 2. General Information of CLDNs section of the manuscript now contains the following sentence: ‘CLDN3 is phosphorylated by Protein Kinase A, which induces disruption of the TJs [26].’ (line 77)
Point 3: In section 3, perhaps an introductory sentence on why gastric cancer is being focused on would help. Does CLDN18 more important in gastric cancer than other cancers?
Response 3: Thank you very much for your helpful suggestion. We agreed with your suggestion and added new sentences of the reason focusing on CLDN18 in GC. We believe that it is very crucial that CLDN18 is specifically expressed in the stomach and GC for searching novel therapeutic targets.
Changes: The 3.1. CLDN18 section of the manuscript now contains the following sentence: ‘The CLDN18 protein is the most studied among the CLDNs, since it is specifically expressed in stomach and GC tissue, compared to other tissues or cancers, which makes it a potential therapeutic target.’ (line 101-103)
Point 4: In section 3.1, please provide an overview of CLDN18 in metastasis since the special issue is on metastasis. Only lymph node metastasis where it did not correlate is mentioned so any literature in metastasis in general would help.
Response 4: Thank you very much for your helpful comment. We agreed with your comment and added new sentences in general and several literatures about the relationship between CLDN18 expression and metastasis.
Changes: The 3.1. CLDN18 section of the manuscript now contains the following sentence: ‘In the relationship between CLDN18 expression and clinicopathological factors in GC, CLDN18 expression was significantly lower in patients with peritoneal metastasis (PM) than those without PM (P =0.01). Meanwhile, CLDN18 expression was significantly higher in patients with bone metastasis than in those without bone metastasis (P =0.01) [45]. Other IHC study showed that CLDN18 expression correlated with lymph node metastasis (P =0.04), high stage disease (III, IV) (P =0.019), and lower incidence of liver metastases (p = 0.009) [46].’ (line 120-126)
Point 5 : Some of the CLDNs sections do not have a mention of metastasis (CLDN 3, 6, 7, 11, 14, and 23). Please check the literature and TCGA for this or mention it is not known in those sections
Response 5: Thank you very much for your constructive suggestion. We agreed with your comment and added new informations and some literatures on metastasis (CLDN 3, 6, 7, 11, 14, and 23).
Changes: The 3.4. CLDN3 section of the manuscript now contains the following sentence: ‘CLDN3 was lower in patients with positive lymphatic invasion, advanced tumor depth [104], and lower TNM stage [105]. Conversely, high CLDN3 expression was related to lymph node metastasis [106].’ (line 113-115)
The 3.6. CLDN6 section of the manuscript now contains the following sentence: ‘In addition, there was no significant relationship between CLDN6 expression and metastatic state (lymph node and organ metastasis) using the TCGA dataset [124].’ (line 157-159)
The 3.7. CLDN7 section of the manuscript now contains the following sentence: ‘There was no significant relationship between CLDN7 expression and metastatic state (lymph node and organ metastasis) [43].’ (line 170-171)
The 3.9. CLDN11 section of the manuscript now contains the following sentence: ‘The relationship between CLDN11 expression in GC and clinicopathological factors correlated with gender, smoking, alcohol consumption, H. pylori infection, and Borrmann classification, not with lymph node metastasis and TNM stage.’ (line 198-200)
The 3.10. CLDN14 section of the manuscript now contains the following sentence: ‘To date, there are limited studies on the expression of CLDN14 in GC; however, evaluation of CLDN14 expression by IHC reported that GC tissues had higher expression compared to proximal normal mucosa and that CLDN14 expression correlated with lymph node metastasis and E-cadherin expression [131].’ (line 209-212)
The 3.12. CLDN23 section of the manuscript now contains the following sentence: ‘Clinicopathologically, CLDN23 was not associated with lymph node metastasis and TNM stage.’ (line 223-224)
Best regards,
Takashi Oshima
Reviewer 2 Report
This review presents novel biological and clinical significance of claudin (CLDN) expression in GC, especially CLDN18, and clinical trials centered around CLDN18.2. It also presents new 11 findings for other CLDNs.
I think this article is informative and well written works.
There are not many reviews that only deal with gastric cancer and claudin. So I think there is an originality.
I recommend addition of working of Claudins in others cancer in general information of CKDNS.
It would be nice to add a table that summarizes the functions of several claudines or their roles in cancer.
Author Response
Response to Reviewer 2 Comments
Point 1: I recommend addition of working of Claudins in others cancer in general information of CLDNs.
Response 1: Thank you very much for your constructive suggestion. We agreed with your suggestion and added working of claudins in others cancer in general information of claudins.
Changes: The 2. General Information of CLDNs section of the manuscript now contains the following sentence: ‘Alternatively, overexpression of CLDNs has also been reported to increase aberrant localization and function in gastric, lung, prostate, ovarian, colorectal and breast cancers, promoting metastasis and progression [7].’ (line 90-92)
Point 2: It would be nice to add a table that summarizes the functions of several claudines or their roles in cancer.
Response 2: Thank you very much for helpful suggestion. We agreed with your suggestion and added new table.
Changes: The manuscript now contains the following new table: ‘Table2. Summary of Claudin’s functions, signaling molecules involved, and clinical significances in GC’ (line 230-231)
Best regards,
Takashi Oshima
Reviewer 3 Report
Dear Author,
The review article "Clauduins and gastric cancer: An overview" by Hashimoto and Oshima was well written. As we all know, high claudin expression has been linked to gastric cancer. As a result, the author emphasized the biological and clinical importance of claudin expression in gastric cancer Overall, the focus of this review article was claudin in gastric cancer.
Minor changes have been made before before publication.
- The author can show a schematic diagram of a major function or pathway.
Author Response
Response to Reviewer 3 Comments
Point 1: Minor changes have been made before publication. The author can show a schematic diagram of a major function or pathway.
Response 1: Thank you very much for your constructive suggestion. We agreed with your suggestion. Although we avoided adding a diagram of the major claudin pathways because there are some parts of the diversity and skepticism in the Claudins pathway, we added a description of the major functions of Claudins to Figure 1.
Changes: The manuscript now contains the following new information as CLDN’s major functions: ‘cell-cell adhesion, barrier function, maintenance of cell porality, selective paracelluler permeability ‘ (Figure 1)
Best regards,
Takashi Oshima